# A Potential Role of IL-6/IL-6R in the Development and Management of Colon Cancer

**DOI:** 10.3390/membranes11050312

**Published:** 2021-04-24

**Authors:** Mimmo Turano, Francesca Cammarota, Francesca Duraturo, Paola Izzo, Marina De Rosa

**Affiliations:** 1Department of Biology, University of Naples Federico II, 80126 Naples, Italy; mimmo.turano@unina.it; 2Department of Molecular Medicine and Medical Biotechnology, University of Naples Federico II, 80131 Naples, Italy; cammarota@ceinge.unina.it (F.C.); francesca.duraturo@unina.it (F.D.); paola.izzo@unina.it (P.I.); 3Ceinge Biotecnologie Avanzate, 80131 Naples, Italy

**Keywords:** IL-6 signaling, membrane IL-6 receptor, soluble IL-6 receptor, colorectal cancer, polyposis syndromes, IL-6-targeting therapy, nutraceutical IL-6 inhibition

## Abstract

Colorectal cancer (CRC) is the third most frequent cancer worldwide and the second greatest cause of cancer deaths. About 75% of all CRCs are sporadic cancers and arise following somatic mutations, while about 10% are hereditary cancers caused by germline mutations in specific genes. Several factors, such as growth factors, cytokines, and genetic or epigenetic alterations in specific oncogenes or tumor-suppressor genes, play a role during the adenoma–carcinoma sequence. Recent studies have reported an increase in interleukin-6 (IL-6) and soluble interleukin-6 receptor (sIL-6R) levels in the sera of patients affected by colon cancer that correlate with the tumor size, suggesting a potential role for IL-6 in colon cancer progression. IL-6 is a pleiotropic cytokine showing both pro- and anti-inflammatory roles. Two different types of IL-6 signaling are known. Classic IL-6 signaling involves the binding of IL-6 to its membrane receptor on the surfaces of target cells; alternatively, IL-6 binds to sIL-6R in a process called IL-6 trans-signaling. The activation of IL-6 trans-signaling by metalloproteinases has been described during colon cancer progression and metastasis, involving a shift from membrane-bound interleukin-6 receptor (IL-6R) expression on the tumor cell surface toward the release of soluble IL-6R. In this review, we aim to shed light on the role of IL-6 signaling pathway alterations in sporadic colorectal cancer and the development of familial polyposis syndrome. Furthermore, we evaluate the possible roles of IL-6 and IL-6R as biomarkers useful in disease follow-up and as potential targets for therapy, such as monoclonal antibodies against IL-6 or IL-6R, or a food-based approach against IL-6.

## 1. Colorectal Cancer (CRC): An Overview

CRC is one of the most frequent cancers worldwide and one of the main causes of cancer-related mortality [1]. The onset of CRC is associated with genetic and environmental factors. In hereditary or familial colorectal cancer, genetic predisposition plays a crucial role in disease onset, and germline inactivating mutations in specific oncogenes or tumor-suppressor genes involved in the physiological turnover of colorectal mucosae could predispose a person to hereditary colorectal cancer syndrome. Furthermore, minor variants in the same genes responsible for hereditary cancers cause familial cancers. However, environmental factors represent the main cause of cancer onset for sporadic forms [2,3]. About 30% of all CRCs are hereditary or familial, while about 75% are sporadic [4]. The first-degree relatives of patients with CRC have a three-fold greater risk of CRC than individuals without familial predisposition [5].

During cancer progression, healthy mucosa accumulates genetic or epigenetic alterations in oncogenes or tumor-suppressor genes, evolving into hyperproliferative mucosa and, later, into early, intermediate, and late adenomas, which, in turn, give rise to in situ carcinomas. Eventually, the accumulation of altering mutations in genes involved in cell–cell adhesion, epithelial-to-mesenchymal transition, and extracellular matrix degradation confer to the cancer cells the ability for distant colonization, generating metastases. This well-known sequence, named the adenoma–carcinoma sequence, is also determined by growth factors and cytokines [6].

Lynch syndrome (LS), also known as non-polyposis colorectal syndrome (HNPCC), is a hereditary disease that accounts for about 1–6% of CRCs. LS predisposes one to CRC onset at a younger age than the general population, usually before 50 years [7,8,9], and to an increase in cancer risk in several regions such as the endometrium, stomach, liver, kidney, and brain, as well as an increased risk of certain types of skin cancer. Germline inactivating mutations in mismatch repair (MMR) genes, such as MLH1, MSH2, MSH6, and PMS2, have been associated with LS.

Polyposis syndromes are a heterogeneous group of hereditary syndromes predisposing to colorectal cancers, characterized by the onset of several polyposes of the colon and rectum, which are classified as adenomatous polyposis and hamartomatous polyposis syndromes. Adenomatous polyposis syndromes include familial adenomatous polyposis (FAP), attenuated FAP (AFAP), and MUTYH-associated polyposis (MAP); hamartomatous polyposis syndromes consist of PTEN hamartoma tumor syndrome (PHTS), juvenile polyposis syndrome (JPS), and Peutz–Jeghers syndrome (PJS). While MAP is an autosomal recessive condition, the other polyposis syndromes, both adenomatous and hamartomatous, are inherited in an autosomal dominant manner [1,6]. FAP is an autosomal dominant condition characterized by the development of hundreds to thousands of adenomatous polyps in the gastrointestinal tract, usually within the second decade of life, which, if untreated, inevitably evolve into carcinomas [10]. Phenotypic variability has been described to be associated with specific gene alterations or the localization of the pathogenic variant on the gene, including attenuated, classic, and aggressive phenotypes, characterized by different numbers of polyps and onset ages. Adenomatous polyposis patients can also develop extraintestinal manifestations, including congenital hypertrophy of the retinal pigment epithelium (CHRPE), polyposis of the upper intestinal tract, desmoid tumors, thyroid tumors, and hepatoblastoma [11]. Biallelic germline mutations of the MutY DNA glycosylase (MUTYH) gene and monoallelic mutations of the adenomatous polyposis coli (APC) gene both cause FAP syndromes. However, mutations in MUTYH result in a less severe phenotype than mutations in APC, characterized by the onset of fewer than 100 adenomas and a late age of disease manifestation [12,13]. Pathogenic germline variants in other genes, such as NTHL1, POLE, and POLD1, are also described in polyposis patients [1]. Although many genes are now known to be associated with the onset of polyposis syndromes when altered, polyposis patients often remain without an identified pathogenic genomic variant [14]. The main genes that, when altered, are involved in the hamartomatous polyposis syndromes are the STK11 serine/threonine kinase, associated with PJS syndrome, and the tumor-suppressor gene PTEN, associated with PHTS syndrome. Other genes suggested to be involved in hamartomatous syndromes are BMPR1A, SDHB, SDHD, SMAD4, AKT1, ENG, and PIK3CA [1,15,16].

It is becoming evident that there is a role for interleukin-6 in inflammation and carcinogenesis, through its downstream transcription factors, such as signal transducer and transcription 3 (STAT3), which stimulates cancer cell proliferation and migration. It has been suggested that the main sources of interleukin-6 (IL-6) production during colorectal cancer progression are the tumor-associated macrophages, mesenchymal stem cells, or colon cancer-associated fibroblasts [17,18]. These data are corroborated by the observations that both membrane and soluble forms of interleukin-6 receptor (IL-6R) are upregulated at the blood serum level in colon cancer patients, and this increase correlates with the cancer size, suggesting a role for IL-6R in colorectal cancer progression. In agreement with this observation, the loss of APC, which is a gene considered to be a gatekeeper of intestinal epithelial turnover, has been described to be associated with the upregulation of IL-6 signaling, which, in turn, activates families of proteins that are often upregulated in CRC, such as the Src family kinases (SFKs), YAP, and STAT3 [19].

## 2. IL-6 Signaling

IL-6 is a pleiotropic cytokine taking on pro- or anti-inflammatory roles, produced by many cell types, such as stromal, hematopoietic, epithelial, and muscle cells. It plays a role in various biological mechanisms, such as immune responses, cell survival, apoptosis, and proliferation [20,21,22,23], by acting in concert with other factors, including heparin-binding epithelial growth factor and hepatocyte growth factor [24,25,26,27,28]. IL-6 also regulates the proliferation of intestinal epithelial cells [24] and is involved in the differentiation of various cells, including mast cells and cardiomyocytes [29]. Furthermore, its anti- and pro-inflammatory properties make it able to modulate the responses to several diseases, including immune diseases (e.g., rheumatoid arthritis); chronic inflammatory diseases; and cancers, such as prostatic and bladder cancers, neurological cancers, and B-cell malignancies [25,30,31,32,33]. However, it is also implicated in Alzheimer’s disease, myocardial infarction, Paget’s disease, and osteoporosis [25].

There are two different ways in which IL-6 signaling can be activated. The first is classic signaling, which involves binding between IL-6 and its membrane receptors, the IL-6R (gp80, α-chain, or CD126) [25], on the surface of target cells. Alternatively, IL-6 binds to a soluble interleukin-6 receptor (sIL-R) (or gp55) in a process called “IL-6 trans-signaling”. As shown in Figure 1, transmembrane metalloproteases are able to cleave membrane IL-6R (mIL-6R) and promote the release of the sIL-6R from membrane receptors localized on the plasma membrane of IL-6 target cells, activating IL-6 trans-signaling [34]. Soluble IL-6R can also be produced by a transcriptional mechanism that generates an alternatively spliced mRNA isoform of the receptor without the region encoding the transmembrane domain, yielding a protein that differs at its COOH terminus by 14 amino acid residues [35,36,37]. This mechanism allows the activation of IL-6 signaling even without IL-6R, when the level of sIL-6R circulating is very high, as observed in several pathological conditions.

Scientific evidence suggests that IL-6 trans-signaling mainly acts in a pro-inflammatory manner, promoting neoplastic transformation, whereas IL-6 classic signaling mainly takes place in regenerative or anti-inflammatory processes (Figure 1) [38]. As shown in Figure 2, IL-6 binding to its receptor (IL-6R or sIL6-R) induces the IL-6/IL-6R complex’s interaction with the ubiquitously expressed gp130 IL-6 transducer (β-chain and CD130). This, in turn, results in gp130 dimerization and phosphorylation and the activation of downstream targets, such as receptor-associated kinases (JAK1, JAK2, and Tyk2), which are eventually responsible for cell proliferation and tumor progression [25]. JAK phosphorylation results in the phosphorylation and activation of the transcription factor signal transducer and activator of transcription 3 (STAT3), which represents a crucial step for cancer transformation and progression through the activation of specific target genes typically involved in neoplastic transformation. Indeed, genes involved in cell survival (Bcl2 and survivin); cell proliferation (c-Myc, cyclin D1, and cyclin B); angiogenesis (HIF-1alpha and VEGF); extracellular matrix degradation (MMP2 and MMP9); cell adhesion (ICAM-1); and inflammation (IL-6, IL-17, IL-23, and Cox2) are among STAT3’s target genes (Figure 1) [38].

## 3. Role of the IL-6 Signaling in CRC 

Since IL-6 is able to activate pro- and anti-inflammatory cell responses, it also shows pro- or anti-neoplastic activity, in a cell-specific manner, that is associated with the presence or loss of IL-6R [25]. Different mechanisms are involved in IL-6’s antitumor activity, such as the promotion of macrophage and lymphokine-activated killer cell antitumor activities, as well as an increase in neutrophils’ cytotoxic effects on tumor cells. IL-6 also sustains cancer cell lysis through the activation of C-reactive protein (CRP) and its binding to the phospholipids of cancer cells, which, in turn, activates the component 1q of the complement system. However, IL-6 signaling often induces cancer progression in both autocrine and paracrine manners (Figure 3). In an autocrine manner, cancer cells upregulate cytokines or their receptors [25], while it is now evident that, in several cancer cells—mainly in solid tumors—cytokines act in a paracrine manner [39]. In agreement with these observations, IL-6, as well as IL-6R, and other proteins and cytokines of the acute phase response such as IL-1 and TNF-alpha are upregulated during tumor progression, and they are elevated in the serum in the late tumor stage, which correlates with the disease severity and outcomes [40,41,42,43]. The main sources of IL-6 in CRC are tumor-associated macrophages, mesenchymal stem cells, and IL-6 released by colon cancer-associated fibroblasts [44]. Experimental and clinical studies demonstrated a clear association of sporadic CRC and inflammation-associated colorectal cancer with IL-6 signaling, although the specific mechanism through which IL-6 plays a role during CRC onset and progression has not been completely clarified.

In vitro studies have demonstrated that IL-6 promotes the growth of epithelial colon cancer cells [38,45,46]. Furthermore, scientific data show the upregulation of serum IL-6 in CRC, which correlates with the tumor size [47,48] but, also, with poor prognosis in metastatic colon cancer patients [49,50] and treatment-refractory carcinomas [51,52]. These observations suggest that the blood serum concentration of IL-6 could be considered to be a diagnostic and prognostic biomarker in colorectal cancer patients that correlates with relapse-free survival and recurrence [44,45,47,53,54]. Furthermore, blood serum IL-6 is upregulated in CRC patients, and tissue IL-6 expression is higher in CRC than in healthy colon mucosa. This increase, again, positively correlates with the tumor TNM stage, invasion, and lymph node metastases and risk of relapse [44,55,56,57,58], while a negative correlation has been observed with tumor histological differentiation. These findings support the observation that a high tumor tissue IL-6 concentration is associated with a poor prognosis in patients with CRC, and it might be a useful predictive marker. It has also been suggested to be a potential therapeutic target in CRC [38,44,57]. Recent literature data have shown that a low level of cancer tissue IL-6 improves the disease-free survival as compared with tumor counterpart tissues expressing higher IL-6. It has been suggested that IL-6 could represent a marker for evaluating the immune status of both PD-L1-negative and PD-L1-positive colorectal cancer patients, which could provide useful information for determining the therapeutic strategies [49]. In agreement with this hypothesis, a previous report demonstrated that the inhibition of IL-6 enhanced the efficacy of anti-PD-L1 treatment in murine models of pancreatic cancer [59].

There are several mechanisms by which IL-6 drives cancer progression. Holmer et al. performed in vitro studies and showed a relationship between the expression of IL-6 and IL-6R in a colon carcinoma cell line and the upregulation of the carcinoembryonic antigen (CEA). They observed that IL-6 classic signaling induced only a weak upregulation of CEACAM5 and CEACAM6; however, IL-6 trans-signaling was able to strongly upregulate the CEA antigens in a STAT3-phosphorylation-dependent manner [60]. This finding is in agreement with the observation that intestinal epithelial cells usually do not express membrane IL-6R (CD126) [38]. It has also been demonstrated that vascular endothelial growth factor receptor 2 (VEGFR2) is under IL-6 control in intestinal epithelial cells. Indeed, during inflammation and tumor conditions, cancer stromal fibroblasts upregulate IL-6 release, which, in turn, induces tumor angiogenesis. Interestingly, IL-6R neutralization negatively regulates angiogenesis and tumor growth through IL-6 signaling downregulation, supporting the hypothesis that considers IL-6 signaling proteins to be useful targets in CRC therapy [17,61] and in several other human diseases [49]. The FRA1 protein is among the downstream targets of the IL-6/STAT3 complex. It is a member of the FOS family transcription factors encoded by the FOS ligand 1 (FOSL1) gene that is involved in CRC progression and aggressiveness through EMT activation [62,63]. IL-6-mediated STAT3 activation upregulates HDAC6 deacetylase, which induces FRA1 Lys-116 deacetylation and FRA1 transcriptional activation, which, in turn, activates the expression of NANOG and other stem cell and EMT-specific proteins, leading to CRC progression, aggressiveness, and metastatic transformation. In agreement with these observations, FRA1 has been found to be highly expressed in multiple cancers and plays a crucial role in cancer transformation, cancer cell motility, cancer cell stemness features, and drug resistance [44,64,65,66,67]. Furthermore, it has recently been suggested that IL-6 could promote colorectal cancer progression through the induction of EMT, metastatic cell spread [68,69], angiogenesis, self-renewal, and drug resistance [35,36,37]. It also could negatively affect the host’s antitumor immunity, altering the activity of dendritic cells and cytotoxic T cells that infiltrate tumor microenvironments [49].

CRC progression is also under the influence of tumor-infiltrating lymphocytes (TILs), whose increase is associated with better cancer prognosis [70]. TGF-beta production by TILs downregulates the expression of T-cell IL-6 and prevents cancer progression. Indeed, the suppression of TGF-beta signaling induces IL-6-dependent colorectal cancer cell growth. Interestingly, trans-signaling represents the principal active mechanism of IL-6-mediated signal transduction in CRC cells. A large subgroup of CRC presents a shift from the membrane-localized IL-6 receptor to the soluble isoform, which is known to be involved in the adherence of CRC cells to the vascular endothelium, inducing metastasis [71]. These findings, again, suggest that sIL6-R and other proteins involved in trans-signaling could represent a good target for CRC therapy, since they are involved in all the stages of CRC development [45]. Recent studies have shown that IL-6 modulates the immune status of the tumor microenvironment in a manner that facilitates the metastatic colonization of colon cancer cells through the alteration of antitumor effector cells [49]. Furthermore, pro-inflammatory cytokines, such as IL-6 and IL-1, induce the activation of the inducible form of prostaglandin H synthase, the COX2 enzyme, providing a scientific explanation for the observed CRC-preventive effects of nonsteroid anti-inflammatory drugs [72]. Il-6 trans-signaling, together with STAT3 activation, is also involved in shifting MSH3 from the nucleus to the cytosol, inhibiting its binding with other mismatch repair proteins. MSH3 is a component of the post-replicative DNA mismatch repair system (MMR) that forms the MutS beta complex by heterodimerization with MSH2. MutS beta, which recognizes insertion–deletion loops larger than 13 nucleotides, binds to DNA mismatches, thereby initiating DNA repair. MSH3 alterations are associated with elevated microsatellite alterations at selected tetranucleotide repeats (EMAST), the most common DNA mismatch repair defect in colorectal cancers, observed in approximately 60% of analyzed cancers [73]. In this context, IL-6 signaling drives the compartmental translocation of MSH3 and the EMAST cancer phenotype.

## 4. Clinical Studies on Drugs Targeting IL-6 Signaling for CRC Therapy

Several types of therapeutics are available, such as anti-IL-6 or anti-IL-6R antibodies, soluble gp130Fc (sgp130Fc), and selective small-molecule JAK inhibitors, that are used in the treatment of human diseases [38,74]. An IL-6 inhibitor, siltuximab, has been studied in a phase II and III clinical trial (NCT00841191); no clinical activity was observed when the drug was used as a monotherapy, but it was well-tolerated [75]. A phase II clinical trial (NCT02119676) investigated the efficacy of ruxolitinib, a JAK inhibitor, in combination with the orally available multikinase inhibitor regorafenib. This study showed that ruxolitinib did not improve the overall survival and progression-free survival in patients with relapsed/refractory metastatic colorectal cancer [76]. Other clinical trials investigating the effects of the other JAK inhibitor molecules are ongoing, and no data are available yet. A phase I clinical trial (NCT00657176), testing the effects of the STAT3 inhibitor OPB-31121, has been concluded and demonstrated that it is a safe and well-tolerated drug showing great anticancer effectiveness [77]. However, a phase II clinical trial (NCT02983578) is still ongoing. These and other studies have suggested that IL-6 monoclonal antibodies alone do not show significant anticancer effectiveness in advanced colon cancer, or in other solid tumors [25]; however, it cannot be excluded that they may be effective in the early stage of the disease.

## 5. Role of IL-6 Signaling in Polyposis Syndromes

The presymptomatic molecular diagnosis of at-risk families and cutting-edge endoscopic techniques have enabled adequate follow-up and cancer prevention in polyposis patients. The management of polyposis patients, who are carriers of APC mutations, requires a colonoscopy every one to two years, from the age of 10–12 years. An annual endoscopic follow-up is, instead, required when a patient already presents gastrointestinal adenomatous polyps. A colectomy is necessary when polyps cannot be removed by endoscopic treatment [4,78]. However, endoscopic surveillance, polypectomy, and prophylactic surgery are all invasive clinical procedures. Chemoprevention represents a very attractive approach although a challenge for clinical and molecular research, and therefore, to date, it has not been a possible therapeutic alternative to offer to patients [79]. An association between chronic inflammation of the intestinal epithelium and the onset of adenomas and CRC is now well-documented [18,80]. Specifically, the overexpression of IL-6 and other pro-inflammatory cytokines increased the polyp number and size in APC(min/+) mice [81]. Furthermore, the loss of the APC gene, which is mutated in polyposis patients and in the majority of sporadic human CRCs, activates gp130-mediated IL-6 signaling, which induces STAT3 activation and, in turn, the activation of target genes involved in the neoplastic transformation of colon and rectal mucosa, such as Src family kinases (SFKs), YAP, and Notch [82]. Cytokine release into the peritoneum and inflammation within the intestinal muscularis have been induced by surgical manipulations and are mainly responsible for the associated organ dysfunction and postoperative discomfort [24]. Several trials have suggested that the recurrence of polyps could be reduced by the use of nonsteroidal anti-inflammatory drugs (NSAIDs), although this drug-associated polyp reduction rarely decreases cancer risk [83]. Furthermore, there are side effects, mainly consisting of cardiological toxicity from long-term treatment with NSAIDs. Therefore, there is a pressing need to develop other drugs that are more effective and better tolerated.

It has recently been demonstrated that interleukin-6 induces a nucleus-to-cytosol shift of hMSH3, leading to DNA mismatch repair defects in colorectal cancer cells [73]. Elevated microsatellite alterations at selected tetranucleotide repeats (EMAST) have been observed in a number of cancers, including colorectal cancers and precancerous conditions, such as gastrointestinal familial hamartomatous polyposis. It has been suggested that IL-6 released during inflammation by tumor epithelial cells and immune cells could cause the hMSH3 shift and EMAST genotype, which has often been associated with poor prognosis in patients with colorectal cancers. Patients with hamartomatous polyposis syndromes (HPS) have an increased risk of colorectal and other cancers. Hamartomatous polyps are classified as benign polyps with a low risk of neoplastic transformation, although the progression of polyps to carcinoma has been observed and the pathogenic mechanisms remain unknown. It has been suggested that hMSH3 alterations could be involved in neoplastic transformation driven by PTEN dysregulation in hamartomatous polyps of PHTS patients [84]. In agreement with this hypothesis, we previously demonstrated cytokine pathway dysregulation in patients with manifestations of the “PTEN hamartoma tumor syndrome”, suggesting that the dysregulation of the PTEN pathway could promote, in non-neoplastic cells from the blood and colon tissues of PHTS patients, cell survival and pro-inflammatory stimulation, which could predispose these patients to the development of multiple cancers [85]. A similar mechanism has recently been suggested for polyp formation in PJS patients. Poffenberger et al. and Ollila et al. [86,87] demonstrated that the heterozygous inactivation of the STK11 gene in hematopoietic T cells and gastric stromal cells was, alone, able to drive hamartomatous polyp formation in PJS patients through a paracrine IL-6 release that activates neoplastic transformation in epithelial gastrointestinal cells.

## 6. Nutritional Approaches Targeting IL-6

A high-calorie diet typical of Western populations, together with a sedentary lifestyle, is considered to be a risk factor for the onset of several diseases, including chronic inflammatory diseases and colorectal cancers [24,88,89]. Efforts have recently been made to investigate alternative biomolecules, such as anthocyanins and phenolic acids, that are linked to a reduction in inflammation and risk for a variety of cancers, including colorectal cancers [24].

Neoplastic transformation is conditioned by several dietary-derived products [90]. The administration of alpha-tocopherol (vitamin E), plant-derived glycoprotein (UDN glycoprotein), and isoliquiritigenin originated from licorice root downregulates IL-6 signaling in several cancers [91]. Observational and experimental data have shown that the Mediterranean diet and a diet that reduces the intake of food that generates inflammation and increases the circulating levels of pro-inflammatory cytokines, such as IL-6, have anti-inflammatory effects in several chronic diseases [92], including polyposis. Under this dietary regimen, FAP patients have reported an improvement in their general physical well-being [93]. Furthermore, several studies, including the Greek-ATTICA study, have demonstrated and reported the effectiveness of a Mediterranean diet in the downregulation of circulating IL-6 [93,94,95,96,97,98]. Accumulating evidence has shown that several forms of vitamin E, such as γ-tocopherol (γT), δ-tocopherol (δT), γ-tocotrienol (γTE), and δ-tocotrienol (δTE), have anti-inflammatory, antioxidant, and cancer-preventive effects, mainly through the inhibition of cyclo-oxygenases, lipo-oxygenases, NF-κB, IL-6, and STAT3, in immune and cancer cells [99].

Four dietary regimens have been investigated in relation to inflammation, i.e., the “fats and processed meat” diet, the “beans, tomatoes, and refined grains” diet, the “whole grain and fruits” diet, and the “vegetable and fish” diet. The first diet, which was enriched in fats, processed meat, salty snacks, and desserts was associated with the overexpression of CRP, IL-6, and homocysteine, whereas the whole grains and fruit diet, which included in its regimen whole grains, fruit, nuts, and green leafy vegetables, and the vegetable and fish diet, including fish and dark-yellow, cruciferous, and other vegetables were negatively associated with specific inflammatory biomarkers, including IL-6 and CRP. Interestingly, these findings were independent of lifestyle, ethnicity, and abdominal waist circumference [100]. These data have also been confirmed by other published studies. Azadbakht et al. and Anderson at al. reported a diet consisting of mainly whole grains, fruits, nuts, and vegetables was more often reported to be effective in decreasing the circulating levels of IL-6 and CRP, independently of the ethnicity of the population studied [100], compared to a diet enriched in sweets and desserts [101,102]. Fiber intake was also correlated with a decrease in IL-6 [103].

## 7. Conclusions

In light of the current knowledge, we believe that IL-6 signaling should be the subject of in-depth research about the molecular basis of hereditary and sporadic CRC onset and progression. Further studies in this field could open new perspectives in molecular biomarkers for cancer and targeted therapy. In our opinion, this knowledge suggests the possibility of a nutraceutical approach, consisting of a specific dietary regimen, such as a diet enriched in vitamin E compounds, to prevent/mitigate disease progression in CRC, mainly in gastrointestinal polyposis patients. In the future, more attention should be paid to the alimentary education of the general population, but especially to the dietary counseling of people at risk of CRC, to prevent the onset of the disease and relapses and, also, to mitigate the side effects related to surgery.

## Figures and Tables

**Figure 1 membranes-11-00312-f001:**
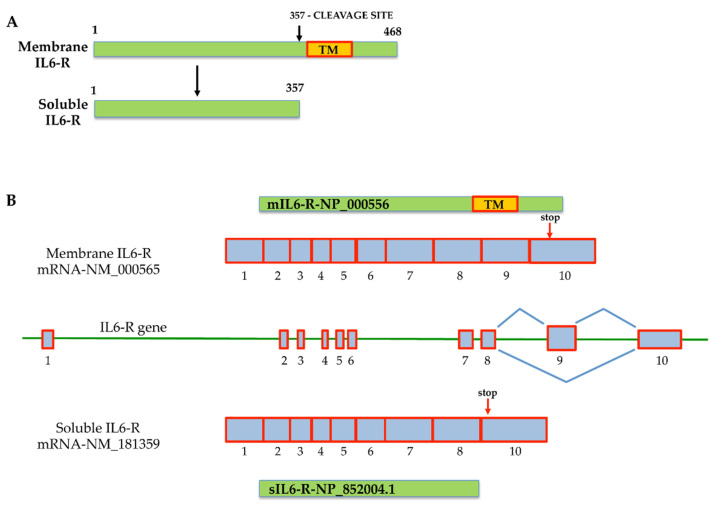
Mechanisms that generate membrane interleukin-6 receptor (mIL-6R) and soluble interleukin-6 receptor (sIL-6R). (**A**) Generation of the sIL-6R protein isoform by metalloprotease cleavage of mIL-6R that removes the transmembrane protein domain; (**B**) generation of the two interleukin-6 receptors (IL-6R) by alternative splicing. The organization of the two alternative splicing isoforms is described. The sIL6-R mRNA is generated by the loss of exon 9, encoding for the transmembrane protein domain (TM). The mIL-6R messenger isoform retains exon 9. The scheme is based on data from NCBI, and the accession numbers of corresponding sequences are reported. Red arrows indicate the stop codon position on exon 10 in both messengers.

**Figure 2 membranes-11-00312-f002:**
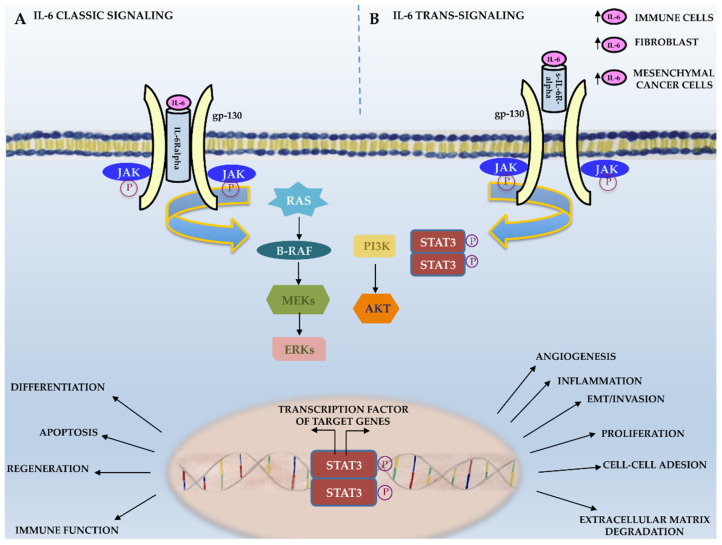
IL-6 signaling. Schematic representation of IL-6 classic and trans-signaling.

**Figure 3 membranes-11-00312-f003:**
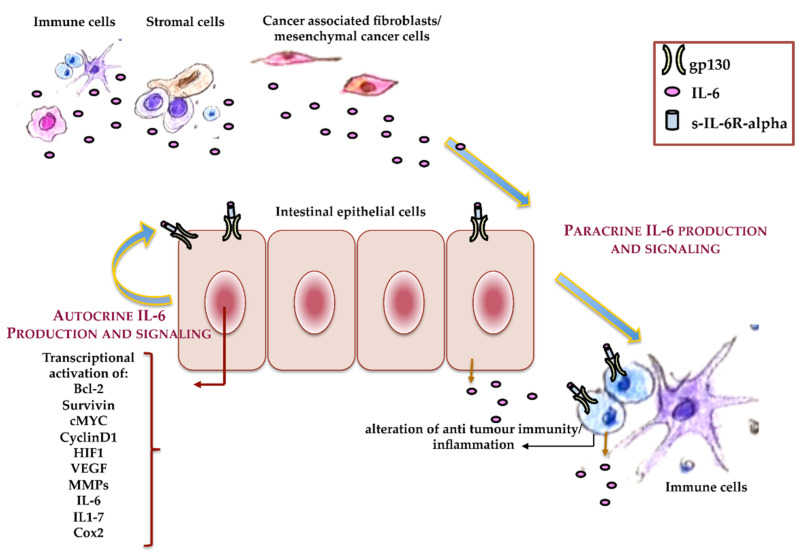
Autocrine and paracrine IL-6. Schematic representation of the roles that autocrine and paracrine IL-6 play during colorectal cancer (CRC) onset and progression.

## Data Availability

Not applicable.

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
