# Peer review of "A Potential Role of IL-6/IL-6R in the Development and Management of Colon Cancer"

_membranes, 2021, doi:10.3390/membranes11050312_

Round 1
Reviewer 1 Report
Authors review the role of cytokine IL6 and its receptor in CRC. Inflammation and CRC has been well studied. Many anti-inflammatory drugs are evaluated for CRC prevention. However many of these agents could not be clinically approved inspite of strong preclinical efficacy due to many side effect with chronic intake. In this context other targets of inflammation need to be validated and studied. IL6 is a well recognized proinflammatory cytokine and its role in CRC is also well studied. This review summarized some of these details however they did not address the anti-IL6 therapies performed in crc patients, existing agents, their efficacy and drawback. Nutrition targeting section is also not very specific to IL6. Authors may do more detailed literature search to include this info as well other wise overall review is well written and focused on the topic selected.
Author Response
Response to Reviewer 1 Comments
Point 1: This review summarized some of these details however they did not address the anti-IL6 therapies performed in crc patients, existing agents, their efficacy and drawback.
Point 2: Nutrition targeting section is also not very specific to IL6. Authors may do more detailed literature search to. 

Response to point 1: We really thank the reviewer for these two interesting observations. As requested, we add in the manuscript a small paragraph (pages 8) entitle “Clinical studies on drug targeting IL-6 signaling for CRC therapy”, that give an overview on the clinical trials performed in CRC with agents targeting molecules of IL-6 signaling.
Response to point 2: We have also revised the paragraph entitle “Nutritional approach targeting IL-6”, trying to highlight, better than before done, the nutritional regimen specific to control IL-6, enriching the contents of this paragraph with numerous other bibliographical references.

Reviewer 2 Report
This manuscript explains accumulating reports regarding IL-6 signaling and CRC development.
However, the manuscript is not suitable for publication at the present form. Although the scientific contents itself is acceptable, the manuscript requires extensive proofreading, with regard to paragraph structure, English editing and so on. These flaws of the manuscript make it hard for readers to understand the contents. Therefore, I recommend major revision.
Author Response
Point 1-2: this manuscript explains accumulating reports regarding IL-6 signaling and CRC development.
However, the manuscript is not suitable for publication at the present form. Although the scientific contents itself is acceptable, the manuscript requires extensive proofreading, with regard to paragraph structure, English editing and so on. These flaws of the manuscript make it hard for readers to understand the contents. Therefore, I recommend major revision.
Response to point 1: Please provide your response for Point 2. (in red)
As suggested by the reviewer, we have revised the paragraph structure as follow:
Paragraph 1, “Colorectal cancer: an overview” is unchanged;
Paragraph 2, “The IL-6 signaling”, is unchanged;
Paragraph 3, that was: “IL-6 in CRC”, now becomes: “Role of the IL-6 signaling in CRC”.
Paragraph 4, entitled “Clinical studies on drug targeting IL-6 signaling for CRC therapy” has been added;
Paragraph 5, “Role of IL-6 signaling in polyposis syndromes”; is the previous Paragraph 4: “Future perspective for polyposis syndrome managements”,
Paragraph 6, “Nutritional approach targeting IL-6”, is the previous Paragraph 5;
Paragraph 7, “Conclusions”, is the previous Paragraph 6.
Response to point 2: Furthermore, we ask to the MDPI Editorial Office for English language revision of the entire manuscript

Round 2
Reviewer 2 Report
The revised manuscript did not reflect the points I indicated at the first round of revision.
1. There are paragraphs consisted of just one sentences. eg. 213,173, 175
2. The terms used within the manuscript are not consistent. IL-6 IL6 Il-6; s-IL-6R-alpha sIL-6R, soluble IL-6R
3. Other numerous editorial and grammatical errors.
Author Response
Response to Reviewer 2 Comments
The revised manuscript did not reflect the points I indicated at the first round of revision.
Point 1: There are paragraphs consisted of just one sentences. eg. 213,173, 175
Response to point 1: We have rearranged the structure of paragraphs within the whole manuscript. All changes are reported in the tracked form of the manuscript uploaded (lines: 40, 43, 67, 76, 82, 84, 94, 124, 143, 145, 170, 173, 178, 186, 192, 199, 212, 219, 226, 244, 253, 256, 276, 287, 305, 309, 311, 329, 340, 349, 363, 381, 390).
Point 2: The terms used within the manuscript are not consistent. IL-6 IL6 Il-6; s-IL-6R-alpha sIL-6R, soluble IL-6R
Response to point 2 We have revised and corrected the terms indicated by the reviewer and made them consistent with all parts of the text
Point 3: Other numerous editorial and grammatical errors.
Response to point 3: An in-depth manuscript editing has been carried out by MDPI's English Editing Services as “Specialist edit”, that includes a check of the grammar, spelling, punctuation, and phrasing followed by a check of the overall structure and clarity of expression.

Round 3
Reviewer 2 Report
I recommend the publication of this manuscript in 'membranes' journal.